



# Structure of mesoscale eddies in the vicinity of Perth Submarine Canyon

Sharani Kodithuwakku, Charitha Pattiaratchi, Simone Cosoli, Yasha Hetzel

5  School of Engineering and the UWA Oceans Institute, The University of Western Australia, 35 Stirling Highway, Perth WA 6009 Australia.

Correspondence to: Sharani Kodithuwakku (sharani.kodithuwakku@research.uwa.edu.au)

**Abstract.** Mesoscale eddies represent discrete, rotating fluid particles that are different compared to their ambient aquatic environment. Understanding the dynamics of mesoscale eddies requires observations, not only of their horizontal structure, such as is available through satellite data, but also of their vertical structure. This study investigates the surface and subsurface characteristics of mesoscale eddies in the vicinity of Perth submarine canyon (30.5–33.5ºS, 112–116ºE) off the southwest coast of Western Australia. Satellite remote sensing (altimetry, temperature, and ocean color) observations were used to understand the surface characteristics while the vertical structure was investigated using ocean glider data collected between 2010 and 2017 through the Integrated Marine Observing Systems (IMOS). Eight Seaglider missions that intersected eddies revealed nine distinct vertical structures, comprising four cyclonic and five anti-cyclonic eddies. Isotherms and isohalines exhibited upwelling in cyclonic eddies, corresponding to mixed layer depth shoaling, and downwelling in anti-cyclonic eddies, aligning with mixed layer depth deepening. Anti-cyclonic eddies exhibited higher surface chlorophyll concentrations than cyclonic eddies, with coastal eddies, regardless of their sense of rotation, displaying elevated surface chlorophyll levels attributed to the entrainment of coastal waters. Offshore eddies featured lower surface chlorophyll concentrations and a distinct subsurface chlorophyll maximum.



## 1 Introduction

Mesoscale eddies (diameter >50km) are isolated ecosystems with unique physical, chemical, and biological properties that are different from their surrounding environment. In oligotrophic environments they are often referred to as oases in the ocean due to localized high productivity (Rennie et al., 2009). Mesoscale eddies play a significant role in the dynamics of eastern boundary currents (EBCs) that include (Chelton et al., 2011): (1) transport of water masses, heat, salt, and nutrients both horizontally and vertically and contribute to the redistribution of properties affecting temperature, salinity, and nutrient concentrations; (2) induce mixing and stirring influencing the vertical and horizontal distribution of properties that influence biological productivity by affecting nutrient availability and phytoplankton distribution; (3) modulate the structure and strength of boundary currents by interacting with the main current flow and can either enhance or weaken currents by altering the momentum balance; and, (4) enhance biological productivity by bringing nutrient-rich waters to the photic zone through upwelling processes that can create favorable conditions for phytoplankton growth.

Majority of eastern ocean boundaries are associated with upwelling, and equatorward surface flow; however, the southeastern boundary of the Indian Ocean is a contrasting scenario where the poleward-flowing Leeuwin Current (LC) flowing against the prevailing equatorward winds promotes downwelling and an oligotrophic environment off Western Australia (WA) (Pattiaratchi & Woo, 2009; Smith et al., 1991). The Perth submarine canyon (31–33ºS, 114–115ºE) intrudes into the continental shelf and influences the shelf circulation, including that of the LC (Rennie et al., 2009; Trotter et al., 2019). In addition, it is a relatively high productivity region along the entire oligotrophic coastline off WA (Pattiaratchi, 2007). The LC is characterized as a warm, low-salinity, and nutrient-poor surface current that flows over the canyon and extends to a depth of 200m. Underneath the LC is the Leeuwin Undercurrent (LUC), which is an equatorward undercurrent transporting high salinity and oxygen waters within its core at 400-600m, interacting strongly with the shelf slope (Woo and Pattiaratchi, 2008).

Although the LC and its eddy field dominate mesoscale circulation off WA understanding of eddy vertical characteristics has been limited by a lack of data from the subsurface (Kodithuwakku et al., 2023). Identifying the vertical structure of these highly energetic rotating structures through traditional field measurements is challenging, given their high temporal and spatial variability. Due to these challenges, the presence of eddies in the Perth submarine canyon has been documented

through satellite altimetry observations and regional modelling investigations (Rennie et al., 2007; 2009b), but few detailed studies have been undertaken. Previous studies off WA identified that anti-cyclonic eddies exhibited higher chlorophyll concentrations than cyclonic eddies (Dufois et al., 2014; Moore et al., 2007), thus motivating further investigation. Recent

advancements in both remotely sensed and in-situ data integrated analysis, when analyzed together, has enabled a fuller understanding of the characteristics of eddies in global oceans (Zhang et al., 2016). It is generally accepted that mesoscale eddies can influence the hydrography of the water column from the surface to 2000 m water depth (Fieux et al., 2005; Morrow et al., 2003). The vertical displacement of warm and cold water is associated with the vertical transport of biogeochemical materials, an occurrence anticipated to exert influence over local ecosystems (He et al., 2019).

In recent years, repeated deployments of ocean gliders through the Integrated Marine Observing System (IMOS) off southwestern Australia (Pattiaratchi et al., 2017) have provided an opportunity to examine the subsurface structure of these mesoscale eddies to fill the knowledge gap and improve our understanding of mesoscale eddies in the region. Globally, ocean gliders have proven to be a versatile and economically efficient complement to ship-based survey data, which are sparse across most of the world's oceans. A number of studies have highlighted the potential for ocean gliders to monitor

subsurface characteristics of mesoscale eddies around the globe, including: in the Labrador sea (Frajka-Williams et al., 2009; Hatun et al., 2007), Algerian basin (Cotroneo et al., 2016), Pacific Solomon sea (Gourdeau et al., 2017), South China Sea (Chen, 2022; Li et al., 2019; 2020; Shu et al., 2019), Mediterranean Sea (Barcelo-Llull et al., 2019), east of Taiwan (Jan et al., 2019), eastern tropical North Atlantic (Kolodziejczyk et al., 2018), Bering Sea (Ladd et al., 2020), Gulf of Alaska (Martin et al., 2009), Gulf of Mexico (Meunier et al., 2018; 2021; Molodtsov et al., 2020; Rudnick et al., 2015; Zhang et al.,

2023), and off the Washington and Canada coasts (Pelland et al., 2013; 2018).

This paper aims to define the vertical structure of mesoscale eddies within the Perth Canyon, using ocean glider and remote sensing data spanning the period 2010−2017. Satellite data were used to provide spatial context to high-resolution vertical profiles intersecting eddies in the vicinity of the Perth submarine canyon, leading to an improved understanding of mesoscale dynamics in the region.




## 1.1 Study region

The study region (30.5–33.5ºS, 112–116ºE) encompasses the Perth submarine canyon (31–33ºS, 114–115ºE) off southwestern Australia (Fig. 1a). Submarine canyons are integral features of continental-shelf margins, holding significant

ecological importance as hubs of biodiversity and drivers of ecosystem dynamics (Trotter et al., 2019). The Perth Canyon descends from the continental shelf break at 200 m down to 6000 m, is steep-sided and narrow, and is a critical feeding habitat for pygmy blue whales during summer, playing a pivotal role in WA's coastal oceanography and endangered species conservation (Rennie et al., 2009a). This region is frequently occupied by mesoscale eddies as identified by eddy detection algorithms using High Frequency Radar data (Bitencourt et al., 2024; Cosoli, et al., 2020) and satellite altimetry (Fig. 1).

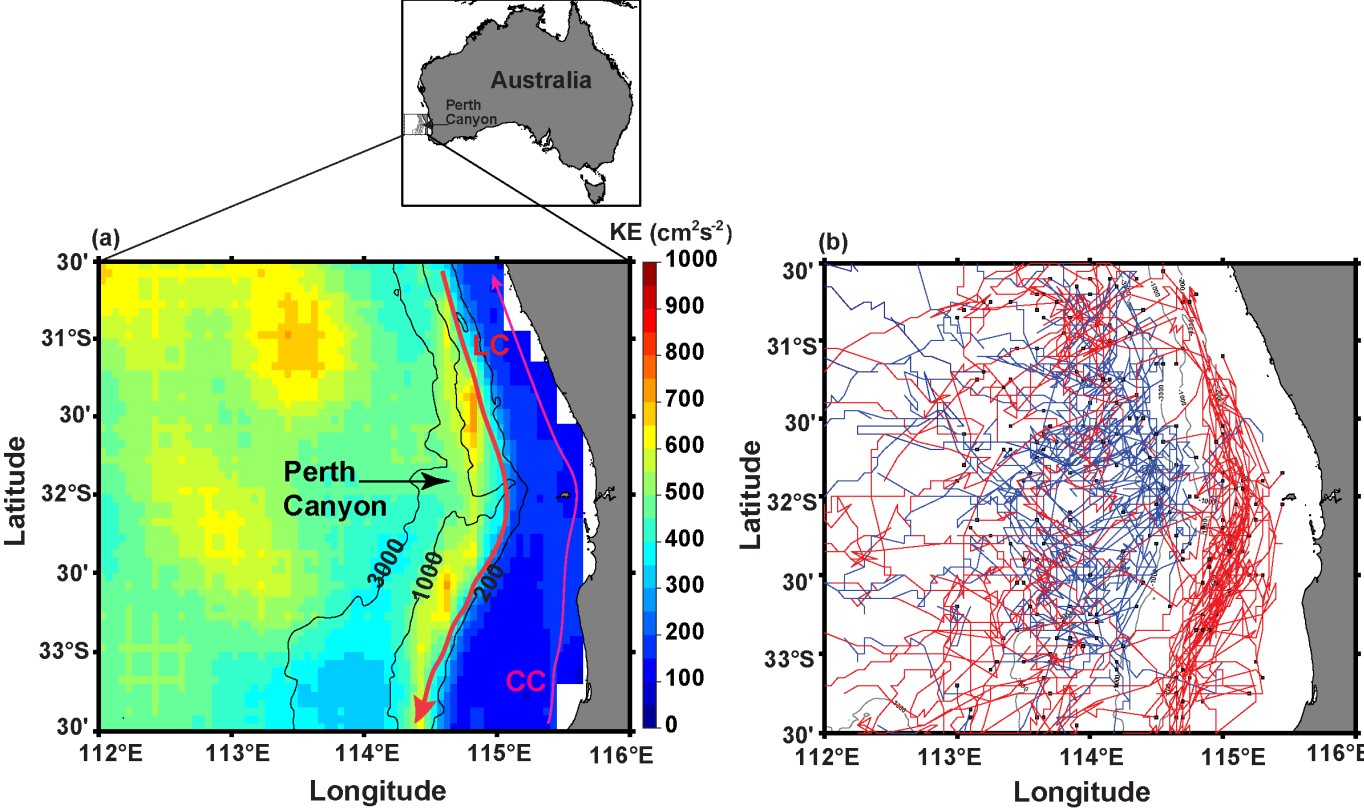


**Figure 1: (a) Mean kinetic energy distribution over 2010−2017, featuring the Perth submarine canyon and its bathymetric contours (in meters). (b) Tracks of cyclonic eddies (solid blue lines) and anti-cyclonic eddies (solid red lines) observed throughout the years 2010 to 2017. The initial positions of each eddy track are indicated by black squares (For further details on the eddy detection algorithm used, refer to section 2.2, based on Nencioli et al. (2010)). The Leeuwin current (LC) shown with a red arrow**

**and the Capes Current (CC) shown with a pink arrow.**



The Perth Canyon is primarily influenced by the Leeuwin Current System, consisting of the Leeuwin Current (LC) in the upper 300 m and the Leeuwin Undercurrent (LU) at depths 300-800m. The LC flows southward along the shelf break and extending over the canyon (Pattiaratchi & Woo, 2009; Woo & Pattiaratchi, 2008). The LC is enhanced by inflow from the southern arm of South Indian Counter Current (sSICC; Wijeratne et al., 2018). Near to the coast, the presence of LC is

evident through downsloping isotherms and isohalines, characterized by the presence of warm and low salinity water in the upper layer (Fieux et al., 2005; Morrow et al., 2003; Wijeratne et al., 2018). LC core, with peak southward velocities exceeding 0.5 ms$^{-1}$ at the surface, is centered between 114.9–115ºE offshore from the 300m isobath (Bitencourt et al., 2024; Cosoli et al., 2020; Feng, 2003; Wijeratne et al., 2018). The Perth Canyon is located in a critical latitude for strong diurnal-inertial resonance with the resonant wind driven inertial currents extend to ~500 m water depth (Mihanović et al., 2016;

Pattiaratchi, 2020). The LU, a persistent subsurface boundary current ranging from 300−800m in depth, flows equatorward flow along the continental slope with peak velocities at 450–550m (Trotter et al., 2019; Wijeratne et al., 2018; Woo & Pattiaratchi, 2008). The upsloping isolines correspond to the equatorward flow of the LU, carrying colder and saltier water (Fieux et al., 2005). Interactions between opposing currents and topographic influences often results in the formation of eddies, particularly in regions where the shelf exhibits curvature (Allen et al., 2001; Rennie, 2005).

The presence of a submarine canyon can significantly impact water flow by causing abrupt changes in depth and isobath direction (Allen et al., 2001; Kämpf, 2012). These changes are expected to modify the behavior of currents, potentially altering vorticity, and inducing mesoscale eddies. Mesoscale eddies often form as a result of the interaction between coastal currents and topographic features, particularly submarine canyons. The LC often exhibits meandering flow that has been linked to the generation of surface eddies (Cosoli et al., 2020). Pygmy blue whales frequenting the Perth Canyon during the

summer reflect high productivity, fostering the production of substantial krill populations. Eddy formations within the canyon further stimulate aggregation, extend nutrient and biota residency, and enhance vertical transport processes (Rennie et al., 2009).

Woo and Pattiaratchi (2008) and Pattiaratchi and Woo (2009) identified five upper Indian Ocean (<1000m) water mass types, aligning with southern Indian Ocean water masses. The primary water mass types and their characteristics within the

Perth Canyon are specified in Table 1. Tropical surface water (TSW) and South Indian central water (SICW) located in the



top 300m and are associated the Leeuwin Current (LC). SICW consists of higher salinity water (Trotter et al., 2019). When the LC is weaker, generally from October to March, there is an absence of TSW and SICW is located at the surface (Woo and Pattiaratchi, 2008). The subantarctic mode water (SAMW) is associated with the Leeuwin Undercurrent and has a local oxygen maximum.


**Table 1. Water Mass types within the Perth canyon**

| Water Mass | Depth | Temperature | Salinity |
| --- | --- | --- | --- |
| Tropical surface water (**TSW**) | 0–100 m | > 22°C | 35.5 |
| South Indian central water (**SICW**) | 100–300 m | 16–22°C | 35.6−35.8 |
| Subantarctic mode water (**SAMW**) | 300–700 m | 8–15°C | 35.4>34.6 |
| Antarctic intermediate Water (**AAIW**) | 700–1000 m | 4.5–7°C | 34.4−34.6 |
| Upper Circumpolar Deep Water (**UCDW**) | 1000–2000 m | < 4.5° | 34.5 |

This table modified from (Pattiaratchi & Woo, 2009; Trotter et al., 2019; Woo & Pattiaratchi, 2008)

## 2   Data and Methods

### 2.1      Satellite remote sensing observations

Satellite Remote Sensing (SRS) data were acquired from the Australian Integrated Marine Observing System (IMOS) via the Australian Ocean Data Network (AODN) portal (https://portal.aodn.org.au/) (Schroeder et al., 2017).

As part of the IMOS SRS facility, the Bureau of Meteorology processed Advanced Very High-Resolution Radiometer (AVHRR) Sea Surface Temperature (SST) data using the algorithms developed by the Group for High Resolution Sea Surface Temperature (GHRSST) format (Beggs et al., 2009; Griffin et al., 2017). These gridded data (L3) are available at

0.02° x 0.02° (~2 km) resolution, "super-collated" L3 file from multiple sensors (L3S), multiple swaths, SSTskin and day+night foundation SST (SSTfnd), daily night-time data from 1992.

(https://thredds.aodn.org.au/thredds/catalog/IMOS/SRS/SST/ghrsst/L3S-1d/ngt/catalog.html).

Ocean color observations (Sea Surface Chlorophyll; SSC) are distributed from NASA satellites (E.g., MODIS AQUA in 2002) with the resolutions of ~1km for MODIS AQUA. The Ocean Color sub-facility within the IMOS SRS offers



comprehensive coverage of the Australasian region daily ocean color products of chlorophyll-a concentration by utilizing

raw data from these NASA satellite missions using the OC3 standard method endorsed by the NASA and applied in the

SeaDAS processing software (Schroeder et al., 2016; 2017). The accuracy of IMOS Ocean Color data is validated against in

situ chlorophyll-a observations compiled by the IMOS Bio-optical Data Base initiative.

(https://thredds.aodn.org.au/thredds/catalog/IMOS/SRS/OC/gridded/aqua/P1D/catalog.html).

IMOS satellite altimetry (Gridded Sea Level Anomaly; GSLA) data and surface geostrophic velocity components were

derived from Ocean Current product provided by IMOS SRS facility (https://oceancurrent.aodn.org.au/). This dataset

combines multiple satellite altimetry data corrected with sea level measurements from tide gauges along the coast. The data

at initial spatial resolution of $0.2º × 0.2º$ (~22km × 22km) were interpolated to $0.05º$ (~5.5km) to improve automated eddy

detection.

(https://thredds.aodn.org.au/thredds/catalog/IMOS/OceanCurrent/GSLA/DM/catalog.html).

During each glider mission (see below), GSLA data were averaged over the designated transect period. Corresponding SST

and ocean color snapshots were selected from cloud-free days within the period of the transect.

Kinetic energy (KE), a measure of current intensity, was calculated along with the relative vorticity (ζ) using the geostrophic

velocities derived from satellite altimetry. Relative vorticity (ζ) was used as a quantification of rotational motion of an eddy.

The following equations were used:

$$KE = \frac{1}{2}(u^2 + v^2) \tag{1}$$

$$\zeta = \frac{\partial v}{\partial x} - \frac{\partial u}{\partial y} \tag{2}$$

Where u and v are the east-west component and north-south components of the geostrophic velocity anomalies, respectively

in the x and y directions. Rossby number $R_o = V/fL \approx \zeta/f$, where $V$ represents the velocity scale, L the length scale, $\zeta$ is the

relative vorticity, $f$ is the Coriolis parameter ($f=2\Omega\sin\phi$, with $\Omega$ Earth's angular speed, $\phi$ the latitude).

## 2.2    Eddy detection in Perth submarine canyon

The Vector Geometry (VG) Eddy Detection and tracking Algorithm, introduced by Nencioli et al. (2010), was applied to

geostrophic velocity data to detect mesoscale eddies in the study area. After identifying an eddy using the VG method, its



center was tracked over time. Identifying eddy centers two centers at successive time steps (t1 and t2) belong to the same

eddy that has moved, the algorithm continues to identify the eddy's track. The algorithm was configured to classify tracked

eddies based on their polarity, distinguishing between cyclonic and anti-cyclonic eddies. Consequently, the algorithm

identified and tracked these eddies to generate cyclonic and anti-cyclonic eddy tracks in the Perth canyon from 2010 to 2017

(Fig. 1b).


## 2.3       Ocean glider observations

Ocean gliders are autonomous underwater vehicles capable of self-propulsion by modulating their buoyancy relative to the

surrounding water to provide a forward momentum (Hanson et al., 2017; Pattiaratchi et al., 2017; Testor et al., 2019). In

addition to their cost-effective nature and proficiency in measuring subsurface characteristics, gliders exhibit the capacity for

uninterrupted data acquisition in diverse weather conditions, facilitating strategic positioning to maximize the efficacy of

profile measurements (IMOS, 2023; Pattiaratchi et al., 2017). IMOS Ocean Glider Facility

(https://anfog.ecm.uwa.edu.au/index.php), have been deploying Seagliders between 2008 and 2017.

Seagliders are designed for remote oceanographic profiling down to 1000 km depth, and 6000 km horizontally over an

extended duration (Eriksen et al., 2001). IMOS Seaglider deployments generally had a duration of 60 days, attaining a

maximum depth of 1000 m (Hanson et al., 2017). The Seagliders were fitted with sensors that included Sea-Bird CTDs for

conductivity (salinity), temperature, and depth measurements, a WET Labs Eco puck optical sensor for chlorophyll-a

fluorescence, chromophoric dissolved organic matter (CDOM), and particle backscatter, in addition to dissolved oxygen

sensors (Sea-Bird SBE 43 for Seagliders). In this study we analyzed temperature, salinity, depth, and chlorophyll-a

fluorescence observations to study the vertical characteristics of mesoscale eddies. Depth−averaged velocity was determined

for each glider by subtracting the GPS-measured displacement at the start and end of each dive from the glider's

displacement through the water, yielding the average water velocity along the glider's track (Rudnick et al., 2015).

IMOS ocean glider data were analyzed using Gliderscope (Version 8), a visualization software developed for NetCDF based

ocean glider data by IMOS (Hanson et al., 2017; Woo, 2021; 2023). This simplified access to NetCDF files, allowing for

versatile graphical visualization and segmented data extraction, such as transects along the glider track as well as



calculations of depth-averaged velocities, depth profile data, and TS diagrams. Further analysis used MATLAB and the

m_map toolbox (MATLAB, 2023; Pawlowicz, 2020). Eight Seaglider missions were included in this study, spanning from

October 2010 to January 2017 (Table 2).

**Table 2. Ocean glider missions used in this study**

| Glider mission number | Glider transects name | Glider mission and total deployment period | Eddy type | Transect period | Eddy duration | Maximum surface current speed (ms$^{-1}$) |
|---|---|---|---|---|---|---|
| **1** | **A–A′** | Perth20101026 26 Oct–29 Dec (SG151) | CE | 27 Nov–01 Dec, 2010 | 07 Nov–31 Dec (55 days) | 0.60 |
| | **B–B′** | | AE | 02–09 Dec, 2010 | 22 Nov–11 Dec (20 days) | 0.72 |
| **2** | **C–C′** | Perth20110626_2 26 Jun – 25 Aug (SG520) | CE | 10–13 Aug, 2011 | 04 Aug–19 Sep (47 days) | 0.53 |
| | **D–D′** | | AE | 21–25 Aug, 2011 | 30 Jul–09 Oct (72 days) | 0.60 |
| **3** | **E–E′** | Leeuwin20130522 22 May – 18 Jul (SG540) | CE | 23–26 Jun, 2013 | 31 May–27 Jun (28 days) | 0.55 |
| **4** | **F–F′** | Leeuwin20140901 01 Sep – 18 Nov (SG540) | CE | 12–16 Nov, 2014 | 26 Oct–31 Dec (67 days) | 0.50 |
| **5** | **G–G′** | Leeuwin20150203 03 Feb – 27 Mar (SG516) | | 18–21 Mar, 2015 | | 0.43 |
| **6** | **H–H′** | Leeuwin20150309 1 09 Mar – 22 May (SG154) | AE | 17–21 Mar, 2015 | 03 Mar–19 Apr (47 days) | 0.40 |
| **7** | **I–I′** | Leeuwin20161013 12 Oct – 09 Dec (SG154) | AE | 17–19 Nov, 2016 | 24 Oct–23 Dec (61 days) | 0.70 |
| **8** | **J–J′** | Leeuwin20170427 27 Apr – 03 Jul (SG516) | AE | 13–17 May, 2017 | 08 May–07 Jun (31 days) | 0.65 |




## 3   Results

Analysis of mean kinetic energy (KE) estimates derived from satellite altimetry over 2010−2017 indicated elevated KE
levels across the 500-1000m isobaths, aligning with regions of strong current intensity associated with the LC along the edge
of the continental shelf (Fig. 1a). Majority of anti-cyclonic eddy tracks were observed to align along the 200m depth contour
closer to the coast, whilst cyclonic eddy tracks were located farther offshore (Fig. 1b). Corresponding glider tracks from
eight Seaglider missions between 2010 to 2017 that intersected these eddies (Table 2) were selected for further analysis (Fig.
2) and are detailed in the following sections.

### 3.1   Seaglider Mission 1 (27 Nov−9 Dec 2010)

Satellite altimetry, SST, and ocean color imagery provided an overview of mesoscale eddy activity during Seaglider Mission
1 (Fig. 3, 4).  An eddy pair was observed in the northern part of the study region, consisting of a cyclonic eddy offshore
(A−A'; 31°S, 112.5°E), with an anti-cyclonic eddy on its eastern flank (B−B'; 31°S, 113.5°E). Strong southward flow was
observed between the two eddies, and both eddies propagated to the west during November and December. Satellite
200   altimetry derived geostrophic velocities indicated the rotational direction of eddies (clockwise/ anti-clockwise) by aligning
with the depth-averaged current directions acquired from the Seaglider (represented by yellow arrows). Relative vorticity
(Rossby number) for the cyclonic and anti-cyclonic eddies calculated using the geostrophic velocities were $-1.62 \times 10^{-5}$ s$^{-1}$
($R_o=0.22$) and $1.09 \times 10^{-5}$ s$^{-1}$ ($R_o=0.15$), respectively. This indicated that the Coriolis force was important in the eddy
dynamics. The cyclonic eddy was characterized lower SST and Surface Chlorophyll Concentration; SCC (Fig. 3a,b,c) at its
205   center, i.e., cold core eddy. During the 55-day lifespan of this cyclonic eddy, the Seaglider bisected its path on 27 Nov,
approximately 20 days after the eddy's formation on 7 Nov. Detailed information on glider and eddy tracks is given in Table
2. The cyclonic eddy reached a maximum diameter of ~152 km on 28 November. The other eddy was characterized by anti-
clockwise rotation with positive GSLA, as a warm core eddy with higher SST and higher surface chlorophyll (B−B'; Fig.
3d,e,f). The lifespan of this eddy was shorter, at 20 days, and the glider crossed it approximately 12 days after its formation,
210   just after passing through the core of the cyclonic eddy. The anti-cyclonic eddy had a maximum diameter of 121 km on 8
December as defined by the eddy tracking algorithm.





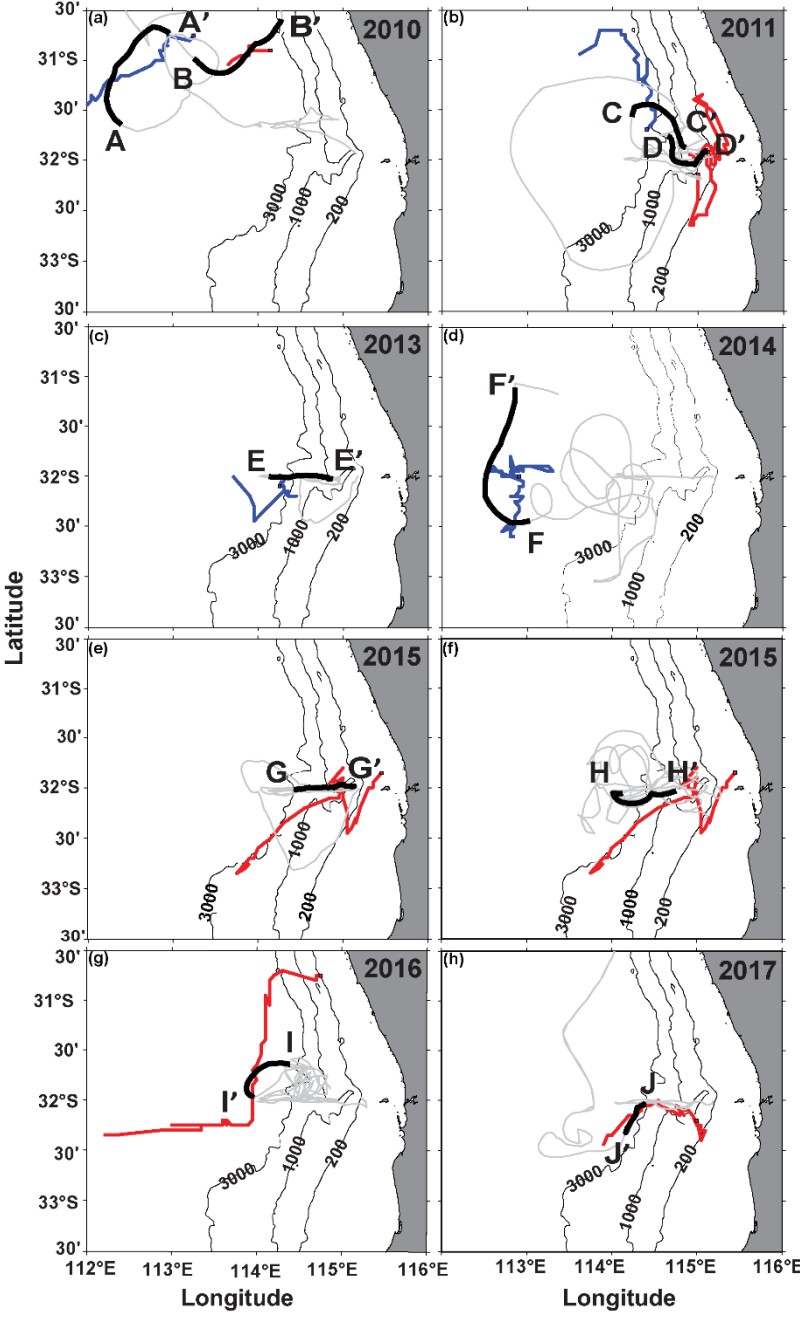

**Figure 2: Selected whole Seaglider trajectories (solid grey lines) showing highlighted track sections glider transects (solid black lines) that intersect selected eddies. The term 'prime' signifies the concluding point of each individual transect. (e.g., A−A′ signifies the glider traversed from A to A′). Corresponding cyclonic eddy tracks are shown with solid blue lines and anti-cyclonic eddy tracks are shown in red (For further details on the eddy detection algorithm used, refer to section 2.2, based on Nencioli et al. (2010)). Bathymetric contours (in meters) and the location of the Perth submarine canyon are also illustrated.**





**Figure 3: Seaglider mission 1 in 2010:** Observed surface characteristics using satellite remote sensing data the Gridded Sea Level Anomaly: GSLA (a, d); Sea Surface Temperature: SST (b, e); ocean colour (c, f) along glider transects A−A' across a cyclonic eddy and B−B' across an anti-cyclonic eddy in the study region. Depth-averaged current directions derived from Seaglider data are indicated by yellow arrows.



**Figure 4: Seaglider mission 1 in 2010: Vertical structure of mesoscale eddies based on Seaglider observations extending to a total depth of 900m and along transects A−A' and B−B' to a depth of 500m for Potential Temperature (a,b,c), Salinity (d,e,f), and Chlorophyll (g,h,i). Chlorophyll data are limited to a depth of 300m only during this mission. Isopycnals (kg/m³) are depicted as white lines. Depth profiles for Potential Temperature (j), Salinity (k), and Chlorophyll (l) are provided up to a depth of 500m. T−S diagrams (m) across glider transects A−A' and B−B' illustrate the water masses: Tropical surface water (TSW), South Indian central water (SICW), Subantarctic mode water (SAMW), Antarctic intermediate Water (AAIW).**



The vertical structure of mesoscale eddies, characterized by potential temperature, salinity, and chlorophyll concentration, was derived from in-situ Seaglider observations. In this glider mission, chlorophyll concentration observations were limited to depths down to 300m, whereas potential temperature and salinity observations extended to a maximum of 900m along the total period of the glider transect. We focus on the water characteristics for depths from the surface to 500 m along the A−A' and B−B' transects (Fig. 4a,d,g).

Transect A−A' within the cyclonic eddy, indicated a shallow mixed layer temperatures > 20°C and higher salinity levels at the surface (>35.6), and lower chlorophyll concentrations (<0.1mg/m$^3$) in the shallow surface layer (<50m) (Fig. 4b,e,h). In contrast, within the anti-cyclonic eddy (along transect B−B'), there was a deeper mixed layer with the 20°C isotherm at 150m depth with warmer temperatures (>20°C), lower salinity levels (<35.5), and relatively higher chlorophyll concentrations (0.2mg/m$^3$) were observed in the near-surface water. There were very different vertical properties between the two eddies. For example, the 20°C isotherm was at the surface at the beginning of the transect within the cyclonic eddy and deepened to 150m water depth within the anti-cyclonic eddy (Fig. 4a). A similar deepening was observed in salinity (Fig. 4d). Here, the mixed layer depth deepened within the anti-cyclonic eddy and shoaled within the cyclonic eddy. The changes in both Potential temperature and salinity between the two eddies extended through the whole depth of the Seaglider observations, to a maxima of 800 m (Fig. 4a,d). A subsurface chlorophyll maximum (1.0 mg/m$^3$ at 80m depth, Fig. 4l) with higher chlorophyll concentration between 75-150m depth was located between 1026−1027 kg/m$^3$ isopycnals in the cyclonic eddy and 1025−1026 kg/m$^3$ in the anti-cyclonic eddy (Fig. 4g). There was a discontinuity between the two eddies, displaying an overlap of isopycnals at 1025 kg/m$^3$ and 1026 kg/m$^3$, along with the subsurface chlorophyll maximum layer (Fig. 4g). There was lower chlorophyll concentration at the surface in the cyclonic eddy whilst there was higher chlorophyll concentration in the whole mixed layer (0-75m) in the anti-cyclonic eddy (Fig. 4h,i). The T−S diagram indicated the presence of the hydrographic characteristics off WA (Fig. 4m), encompassing the water masses TSW, SICW, SAMW, and AAIW through the vertical (Table 1). However, only the anti-cyclonic eddy contained TSW water mass at the surface, whilst in the cyclonic eddy, the surface water consisted of SICW due to the upwelling of the sub-surface water.



## 3.2        Seaglider Mission 2 (10–25 Aug 2011)

Similar to the Seaglider mission 1, Seaglider mission 2, also sampled adjacent cyclonic and anti-cyclonic eddies throughout the duration of the glider transect from 10 to 25 August 2011 (Fig. 5, 6). The satellite imagery along the C−C' transect

(31.5°S, 114.5°E) revealed the presence of a cyclonic eddy, characterized by a lower SST and lower SCC (Fig. 5a,b,c). The lifespan of cyclonic eddy was 47 days, and the glider crossed it approximately 6 days after its formation. Similarly, along transect D−D' (32°S, 115°E) an anti-cyclonic eddy with higher SST and SCC, was present (Fig. 5d,e,f). The lifespan of cyclonic eddy was 72 days, and the glider crossed it approximately 23 days after its formation. The observed cyclonic and anti-cyclonic rotation were in agreement with the depth-averaged current directions from Seaglider data (Fig. 5a,d). The

relative vorticity: $-1.37 \times 10^{-5} s^{-1}$ and $1.32 \times 10^{-5} s^{-1}$, Rossby numbers: $R_o=0.18$ and $R_o=0.17$, and similar diameters: 94km (13 Aug) and 92 km (24 Aug) for the cyclonic and anti-cyclonic eddies, respectively.

The Seaglider mission 2 effectively differentiated between cyclonic eddy and anti-cyclonic eddies. In the cyclonic eddy (transect C−C') the 20°C isotherm and the 35.5 isohaline were located at the surface (Fig. 6b,e). In the anti-cyclonic eddy (transect D−D') these contours were located at depths 130m (temperature) and 175m (salinity) respectively (Fig. 6c,f,i). The

maximum chlorophyll concentrations were at the surface in cyclonic eddy (Fig. 6g) and a subsurface chlorophyll maximum at 50m water depth in the anti-cyclonic eddy (Fig. 6l). As an example, the 18°C isotherm deepened to depths between 50m and 230m within the anti-cyclonic eddy, whilst it shoaled to around 50m within the cyclonic eddy (Fig. 6b, c). The salinity maximum (>35.6) exhibited depth variations, ranging from 50m to 130m in the cyclonic eddy and 220m to 350m in the anti-cyclonic eddy (Fig. 6d,e,f). The salinity changes were significant to water depths > 500m (Fig. 6k).

The water masses were found to align with the characteristic features typically depicted in a T−S diagram off WA (Fig. 6m). Specifically, the surface TSW and SICW masses were associated with the LC.





**Figure 5: As Figure 3 but for Seaglider mission 2 in 2011. Transects C−C' across a cyclonic eddy and D−D' across an anti-cyclonic eddy.**






Figure 6: As Figure 4 but for Seaglider mission 2 in 2011. Transects C−C' across a cyclonic eddy and D−D' across an anti-cyclonic eddy.



### 3.3 Seaglider Mission 3 (23–26 Jun 2013)

Seaglider mission 3 provided evidence for the presence of a cyclonic eddy along the glider transect E−E' from 23 to 26 June
2013 (Fig. 7). In 2013, satellite observations along the E−E' transect (32°S, 114.8°E) identified an eddy with colder SST,
higher surface chlorophyll, and negative GSLA (Fig. 7a,b,c). The rotation pattern was consistent with depth-averaged current
directions derived from Seaglider data (Fig. 7a). During the 28-day lifespan of the cyclonic eddy, Seaglider crossed its path
on 24-25 June, approximately 24 days after the eddy's formation on 31 May. The cyclonic eddy was characterized by a
relative vorticity $-0.73 \times 10^{-5}$ $s^{-1}$ ($R_o$=0.09) and attained a maximum diameter of 100km (25 Jun) within the glider transect
period.

Analysis of vertical structure obtained during glider mission 3 within the cyclonic eddy along transect E−E' revealed
different characteristics compared to, for example Mission 2 (Fig. 5, 6). The isotherms and isohalines were almost horizontal
with downwelling closer to the coast a typical feature associated with the downwelling Leeuwin Current (Fig. 7d,e). The
major feature is higher chlorophyll concentrations ($0.6 mg/m^3$) in the surface layer to ~100m water depth, along the
$1026 kg/m^3$ isopycnal (Fig. 7f). The water masses were different to the typical characteristics represented in a T−S diagram
off WA (Fig. 7j) with the absence of a subsurface salinity maxima associated with the SICW with the higher salinity
occupying the surface layer to 100m water depth (Fig. 7h).





**Figure 7: Seaglider mission 3 in 2013: Observed surface characteristics using satellite remote sensing data the Gridded Sea Level Anomaly: GSLA (a); Sea Surface Temperature: SST (b); ocean colour (c) along glider transects E−E' across a cyclonic eddy in the study region. Depth-averaged current directions derived from Seaglider data are indicated by yellow arrows. Vertical structure of mesoscale eddies based on Seaglider observations along transects E−E' to a depth of 500m for Potential Temperature (d), Salinity (e), and Chlorophyll (f). Isopycnals (kg/m³) are depicted as white lines. Depth profiles for Potential Temperature (g), Salinity (h), and Chlorophyll (i) are provided up to a depth of 500m. T−S diagrams (j) across glider transect E−E' illustrate the water masses: Tropical surface water (TSW), South Indian central water (SICW), Subantarctic mode water (SAMW), Antarctic intermediate Water (AAIW).**



### 3.4     Seaglider Mission 4 (12–16 Nov 2014)

Seaglider mission 4, sampled a cyclonic eddy along transect F−F' located offshore (32°S, 113°E) from 12 to 16 Nov 2014 (Fig. 8). The Seaglider transect traversed the periphery of the eddy from the south to the west and then to the northern periphery of the eddy (Fig. 8a). This cyclonic eddy was characterized by a decrease in SST and a notable reduction in surface chlorophyll concentration. The cyclonic eddy aligned with the depth-averaged current directions derived from Seaglider data, consistent with previous glider missions in this study (Fig. 8a). During the 67-day lifespan of the cyclonic eddy,

Seaglider crossed its path on 12 Nov, approximately 18 days after the eddy's formation on 26 Oct. The cyclonic eddy displayed a maximum diameter of 110 km (12 Nov) and a relative vorticity value of $-0.39 \times 10^{-5} s^{-1}$ ($R_o$=0.05).

Glider mission 4, within the cyclonic eddy along transect F−F', depicted elevated isotherms and isohalines, suggesting lower temperatures (<20°C), higher salinity (>35.6), and a notable decline in chlorophyll concentrations (<0.2mg/m³) in the uppermost layer (<50m) of the water column (Fig. 8d,e,f). The salinity maximum (>35.6) displayed varying depths,

extending from the surface to 110m at the F−F' start of transect and from the surface to 80m at the end of the transect (Fig. 8e). Within this cyclonic eddy, a distinct subsurface chlorophyll maximum was evident, with elevated chlorophyll concentrations (0.8mg/m³) observed between 50m and 150m, coinciding in between the 1026 kg/m³ and 1027 kg/m³ isopycnals (Fig. 8f). When compared with findings from other glider missions conducted within this study, the observed water masses exhibited an absence of TSW at the surface, primarily due to this cyclonic eddy was formed and located further

offshore. Instead, there was a dominance of SICW at the surface. The characteristics of the other two water masses aligned with typical T−S diagram off WA (Fig. 8j).





Figure 8: As Figure 7 but for Seaglider mission 4 in 2014. Transects F−F' across a cyclonic eddy.



### 3.5 Seaglider Mission 5 (18–21 Mar 2015; February mission)

Seaglider mission 5, from 18 to 21 March in 2015, travelled through an anti-cyclonic eddy with a diameter of ~90km along

transect G−G' and had relative vorticity of $0.64 \times 10^{-5}$ s$^{-1}$ ($R_o$=0.08) (Fig. 9). During the 47-day lifespan of the anti-cyclonic

eddy, Seaglider crossed its path on 18 March, approximately 15 days after the eddy's formation on 03 March.

The Seaglider data indicated a uniform surface mixed layer to depth up to 150m depth in both temperature and salinity (Fig.

9d.e). Closer to the surface (<50m), there were warmer temperatures (>23°C), and salinity levels (~35.5). A distinct

subsurface chlorophyll maximum (1.0mg/m$^3$) was observed between 40m to 80m along the 1025kg/m³ isopycnal (Fig. 9f,i).

The water masses exhibited characteristics consistent with the typical features found in a T−S diagram off WA, with

dominance of the TSW mass at the surface and SICW in the sub-surface (Fig. 9j).



**Figure 9: As Figure 7 but for Seaglider mission 5 in 2015 February. Transects G−G' across an anti-cyclonic eddy.**



### 3.6 Seaglider Mission 6 (17–21 Mar 2015; March mission)

Seaglider mission 6, conducted from 17 to 21 March 2015, sampled the same anti-clockwise eddy as in mission 5 (Fig. 9) but this was a separate instrument (Fig. 10). The transect was slightly to the west and sampled from the periphery of the eddy to its center along transect H−H' (Fig. 10a). Details of eddy characteristics are in section 3.5.

The vertical structure of the eddy showed very different pattern to that in Mission 5 that sampled the center of the eddy (Fig. 9).  There were significant changes in the upper ocean temperature and salinity structure. At the beginning of the transect, in

the edge of the eddy with strong currents the 20°C isotherm was located at 70m depth and decreased to 200m at the end of the transect (Fig. 10d). Similarly, the 35.7 isohaline deepened from 50m to 180m (Fig. 10c). A subsurface chlorophyll maximum was present between 50m and 100m, situated along the ~1025kg/m³ isopycnal (Fig. 10f). An interesting feature was changed in the vertical distribution of chlorophyll concentrations across the transect. Approximately at the midpoint of the transect, chlorophyll concentrations extended from the surface to 130m depth (Fig. 10f). This transition coincided with

the maximum changes in the rate of change of Potential temperature and salinity with depth (Fig. 10d,e,f). It also coincided with the Seaglider crossing a filament of chlorophyll extending from the coast to offshore (Fig. 10c). The T−S diagram was similar to mission 6 aligned with characteristic features of a typical T−S diagram for WA waters, with dominance of the TSW mass at the surface (Fig. 10j).





Figure 10: As Figure 7 but for Seaglider mission 6 in 2015 March. Transects H−H' across an anti-cyclonic eddy.



### 3.7 Seaglider Mission 7 (17–19 Nov 2016)

Seaglider mission 7, from 17 to 19 Nov in 2016, documented a large anti-cyclonic eddy, of diameter 165 km, along glider transect I−I' with relative vorticity of $0.4 \times 10^{-5}$ $s^{-1}$ ($R_o$=0.05) (Fig. 11). The transect crossed from the north-eastern periphery to the center of the eddy (Fig. 11a). During the 61-day lifespan of the anti-cyclonic eddy, Seaglider crossed its path on 17 November, approximately 25 days after the eddy's formation on 24 October.

Vertical structure data from glider mission 7, along the flank of the eddy, indicated sloping isotherms and isohalines with the isotherms deepening from 90m to 170 m (Fig. 11d,e). At the beginning of the salinity maximum (>35.7), corresponding to SICW was at the surface and deepened to 100m (Fig. 11e). The subsurface chlorophyll maximum (0.6mg/m$^3$) was present at 90m mainly along the 1026kg/m³ isopycnal (Fig. 11f,i). The water masses of T−S diagram align with the characteristic features typically illustrated in a T−S diagram for WA waters (Fig. 11j).





**Figure 11: As Figure 7 but for Seaglider mission 7 in 2016. Transects I−I' across an anti-cyclonic eddy.**




### 3.8 Seaglider Mission 8 (13–17 May 2017)

Seaglider mission 8, sampled an anti-cyclonic eddy along transect J-J' from 13 to 17 May 2017, spanning its western periphery (Fig. 12). The transect started approximately 5 days after the eddy's formation on 08 May. The anti-cyclonic eddy had a diameter of 115km (17 May) had a relative vorticity value of $0.59 \times 10^{-5}$ s$^{-1}$ ($R_o = 0.08$).

The Seaglider data indicated an almost horizontal 20°C isotherm with the sub-surface salinity maximum at ~150m depth and higher chlorophyll in the top 100 m (Fig. 12d,e,f). There was no subsurface chlorophyll maximum; however, the chlorophyll concentration was significantly higher (>0.8 mg/m³) in the top 100m of the water column along the 1025 kg/m³ isopycnal (Fig. 12f). This coincided with the sampling of higher chlorophyll water from the shelf. The water masses observed aligned with the characteristic features typically depicted in a T−S diagram off WA with lower salinity (<35.3) in the surface layer

(Fig. 12j).







**Figure 12: As Figure 7 but for Seaglider mission 8 in 2017. Transects J−J' across an anti-cyclonic eddy.**



## 4. Discussion

The eastern Indian ocean basin although located in a region of strong southerly winds that promote upwelling is oligotrophic due to the downwelling favorable southward flowing boundary current, the Leeuwin Current (LC). The region includes the most energetic mesoscale eddy field in eastern ocean basins (Pattiaratchi and Siji, 2020; Kodithuwakku et al., 2023). The generation of cyclonic and anti-cyclonic eddies are dependent on the generation of vorticity either side of the LC with anti-cyclonic eddies preferentially generated inshore of the LC and cyclonic eddies offshore (Kodithuwakku et al., 2023). The

Perth canyon is also a local high productivity region supporting a diverse ecosystem dominated by the presence of a large pygmy blue whale population (Rennie et al. 2009a, Trotter et al., 2019). The frequent presence of both mesoscale and sub-mesoscale eddies is a feature in the vicinity of the canyon (Bitencourt et al., 2024; Cosoli et al., 2020, Pattiaratchi, 2007; Rennie et al., 2009b). We identified surface and subsurface characteristics of mesoscale eddies, derived from a combined analysis of data collected from both Seagliders and satellite remote sensing in the Perth submarine canyon (30.5–33.5ºS,

112–116ºE) over the time 2010−2017. Across eight Seaglider missions, a total of ten cross-section transects were examined that included five each of cyclonic and anti-cyclonic eddies. In the study area, there were significant seasonal changes in the ocean dynamics and the Seaglider missions included both summer (Mission numbers 1, 4, 5, 6, 7; Table 2) and winter (Mission numbers 2, 3, 8) deployments. The sampled eddies had diameters that ranged between 90 km and 165 km with Rossby numbers $R_o$=0.05 and Ro=0.22.

Seagliders are driven by buoyancy and as such are challenging to pilot in a straight line in strong currents that are typical of those in the vicinity of eddies. Therefore a 'perfect' cross-section across an eddy was not possible. In this study, we were fortunate that the Seaglider transited from cyclonic to anti-cyclonic eddies in 2 instances (Mission 1: Fig. 3, 4 and Mission 2: Fig. 5, 6). All of the others, the Seaglider sampled the flanks of the eddies (Fig. 7 to 12) but provided valuable insights to changes in the vertical distribution of water properties.

Mesoscale eddies are ubiquitous features of ocean circulation that modulate the supply of nutrients to the photic zone (Dufois et al., 2014). It is generally accepted that there is upwelling at the center of cyclonic eddy that transport colder, nutrient rich water enhancing phytoplankton production. These 'cold' core eddies, due to colder water at the center have shallow mixed layers. In contrast, anti-cyclonic eddies are associated with downwelling at the center ('warm' core eddies)



and deeper mixed layers. The results of this study were in agreement with this paradigm in terms of the mixed layers. In all

the missions, in general, the surface layer (<50m), was warmer (>20°C) and of lower salinity (<35.4) in anti-cyclonic eddies and colder (<20°C), higher salinity (>35.5) water in cyclonic eddies. Isotherms and isohalines displayed an upwelling pattern within cyclonic eddies, coinciding with shoaling of the mixed layer depth, whilst they exhibited a depressed pattern within anti-cyclonic eddies, corresponding to a deepening of the mixed layer depth. Within the anti-cyclonic eddies, the depressed isotherms and isohalines were found at depths ~150m−250m with warmer temperature (>22°C) and higher salinity (>35.7)

waters. The down sloping of the isolines is a distinctive feature of anti-cyclonic eddies (Feng et al., 2007; Fieux et al., 2005; Rennie et al., 2007) and this feature was present in all Seaglider missions that sampled anti-cyclonic eddies (Table 2). A notable feature of the rapid changes in the depths of isotherms and isohalines between adjacent cyclonic and anti-cyclonic eddies that were sampled in Missions 1 and 2. Here, depth changes of 180m (Mission 1) and 150 m (Mission 2) between the two types of eddies were recorded which is a significant change.

In terms of chlorophyll concentrations, the results did not follow the usual paradigm in that cyclonic eddies associated with upwelling did not indicate higher chlorophyll concentrations at the center of the eddies. The data collected during the summer months (Mission numbers 1, 4, 5, 6, 7; Table 2) all indicated the presence of a deep chlorophyll maximum and those collected during winter (Mission numbers 2, 3, 8) indicated higher chlorophyll concentrations at the surface. Also, it was observed that at the surface, chlorophyll concentrations were higher in anti-cyclonic eddies compared to cyclonic eddies.

Additionally, both cyclonic and anti-cyclonic eddies located closer to the coast exhibited higher surface chlorophyll levels. In contrast, offshore eddies west of 114ºE displayed lower surface chlorophyll concentrations but exhibited a distinct deep chlorophyll maximum. The deep chlorophyll maximum is a common feature along this coast particularly in the summer months (Hanson et al., 2004; Chen et al., 2019). This has been attributed to relatively clear water in the upper ocean and oligotrophic conditions. The deeper photic layer allows for light penetration to deeper water where there is a supply from

nutrients from below the pycnocline that allows for the larger celled phytoplankton to grow (Hanson et al., 2004). The assumption that anti-cyclonic eddies contain lower chlorophyl concentrations were not observed. Although the anti-cyclonic eddies indicated deep mixed layers associated with downwelling, the higher chlorophyll concentrations within the eddies were as a result of entrainment of chlorophyll-rich shelf waters by anti-cyclonic eddies forming near the shelf, which were



then transported offshore as they moved westward (Moore et al., 2007; Waite et al., 2007; Dufois et al., 2014; Gaube et al.,

2013). The entrainment of higher chlorophyll from the shelf in the study area can be illustrated through the use of satellite

derived surface chlorophyll concentrations recorded in 2012, 2015, and 2017. These correspond to with the observation of

analogous anti-cyclonic eddies during specific Seaglider missions (Missions 5-6 in 2015 and Mission 8 in 2017). The

entrainment of highly productive water from the shelf by these anti-cyclonic eddies was impeded during our glider missions

due to cloud cover. (Fig. 13).

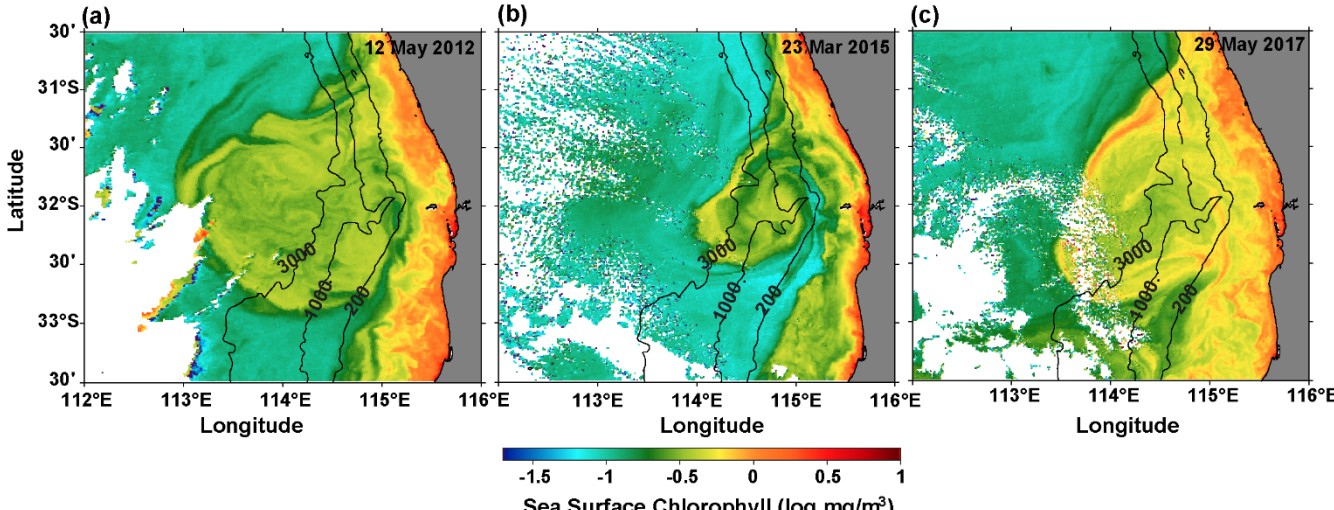


**Figure 13: Snapshots of satellite-derived, cloud-free ocean colour distributions indicate the entrainment of higher chlorophyll water from the shelf within the eddy.**

Previous studies have reported that anti-cyclonic eddies in the southeast Indian Ocean demonstrated a preference for

entraining water from the continental shelf, characterized by heightened levels of both phytoplankton biomass and nutrients

(Gaube et al., 2013; Pattiaratchi, 2017). The results from this study confirmed these findings. As expected, both cyclonic and

anti-cyclonic eddies reflected upwelling and downwelling within the core of the eddies through shallow and deeper mixed

layer depths as would be expected in Coriolis force dominating the dynamics with Rossby numbers ($R_o$) ~0.05 to 0.22.

However, chlorophyll concentrations an indicator of primary productivity were higher in the surface ocean in anti-cyclonic

eddies. This was due to the entrainment of high SSC from the continental shelf during the formation of the eddies (Fig. 14)

and facilitated by the dominant generation of anti-cyclonic eddies inshore of the LC closer to the coast (Kodithuwakku et al.,

2023).



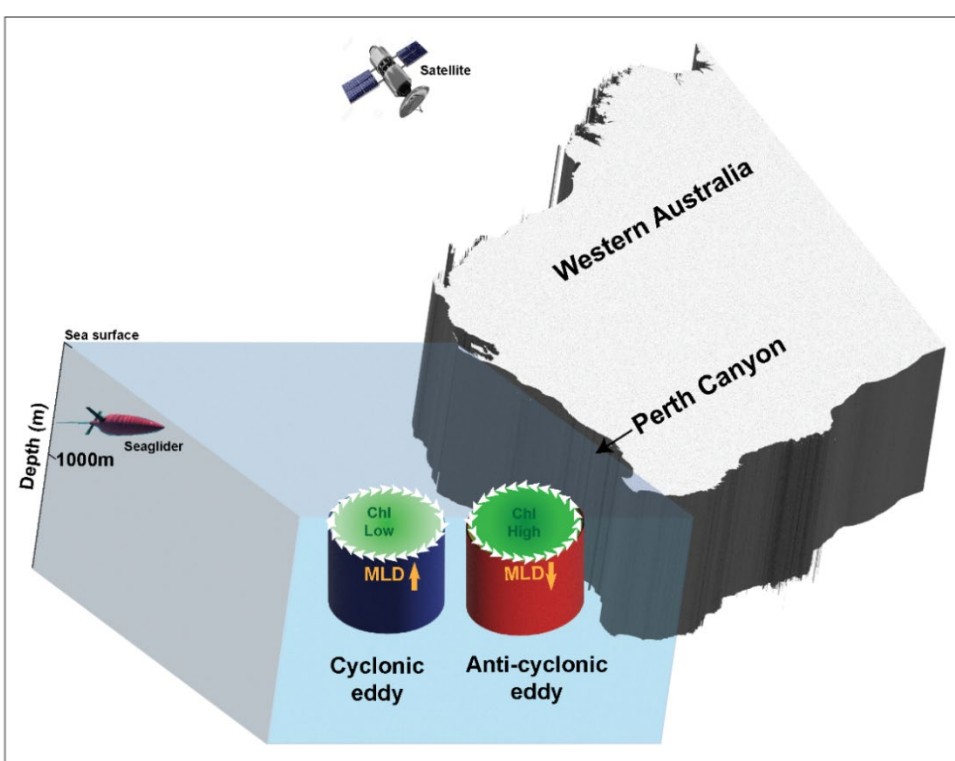

**Figure 14: A schematic diagram illustrating the surface and subsurface characteristics of mesoscale eddies observed through**
**satellite remote sensing and Seaglider missions. Note that objects are not to scale. (MLD- Mixed Layer Depth, Chl- Chlorophyll,**
**SST- Sea Surface Temperature.)**



## 5. Conclusions

Seaglider observations and satellite remote sensing data over the period 2010−2017 were used to analyze the vertical and horizontal structures of nine mesoscale eddies with the poleward−flowing eastern boundary current Leeuwin Current in the Indian Ocean off the southwest coast of Western Australia. The study area also included the Perth canyon. The mesoscale eddies in the Perth submarine canyon exhibited the following characteristics:

1. The data analysis included 8 Seaglider missions with both cyclonic and anti-cyclonic eddies. The eddy diameters were ranged from 94-152 km and 90-165 km for cyclonic and anti-cyclonic eddies respectively. The maximum currents in the eddies were up to 0.72 ms$^{-1}$. The Rossby numbers ($R_o$) were ranged from 0.05 and 0.22.

2. Cyclonic eddies contained shallow mixed layers and anti-cyclonic eddies deep mixed layers. In all the missions, the surface layer was warmer (>20°C) and of lower salinity (<35.4) in anti-cyclonic eddies and colder (<20°C), higher salinity (>35.5) water in cyclonic eddies.

3. There were rapid changes in isotherm and isohaline depths between adjacent cyclonic and anti-cyclonic eddies with vertical changes up to 180m between the two types of eddies.

4. A distinct subsurface chlorophyll maximum was common in both types of eddies during the summer months.

5. Anti-cyclonic eddies exhibited higher surface chlorophyll concentrations than cyclonic eddies due to the entrainment of higher chlorophyll water from the continental shelf.

**Data Availability**

All the data used in this study are publicly available. Data are available from https://portal.aodn.org.au/, part of the Australian Integrated Marine Observing System (IMOS), which is enabled by the National Collaborative Research Infrastructure Strategy (NCRIS). All figures were generated using Matlab software from Mathworks, Inc (http://www.mathworks.com), R2022b.

**Author Contributions**

This study was done as a part of PhD research by **S. K.** All data analysis were done by **S. K.** with the supervision of **C. P.**, **S. C.**, and **Y. H.** All authors have read and agreed to the published version of the manuscript.

**Conflict of Interest**

The authors declare that the research was conducted in the absence of any commercial or financial relationships that could be construed as a potential conflict of interest.

**Acknowledgments**

The data were sourced from the Australian integrated marine observing system (IMOS) that is enabled by the National collaborative research infrastructure strategy (NCRIS). It is operated by a consortium of institutions as an unincorporated joint venture, with the University of Tasmania as the lead agent. We would like to thank the IMOS ANFOG Ocean Gliders Facility operated by the UWA ocean glider team. Sea Surface Temperature (HRPT AVHRR SSTskin) retrievals were produced by the Australian Bureau of Meteorology as a contribution to the IMOS. The satellite imagery were acquired from NOAA spacecraft by the Bureau, Australian Institute of Marine Science, Australian Commonwealth Scientific and Industrial Research Organization, Geoscience Australia, and Western Australian Satellite Technology and Applications Consortium.

**Financial support**

This research is part of PhD research by Sharani Kodithuwakku that was funded by a scholarship for international research fees (SIRF), University postgraduate award research priorities fund (UPARPF). And a top-up scholarship by the Western Australian marine science institution (WAMSI) under the Western Australian satellite technology and applications consortium (WASTAC) small grants.



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
