# Peer review of "Structure of mesoscale eddies in the vicinity of Perth Submarine Canyon"

_EGUsphere, 2024_

## Referee Comment (RC1)

Dear Editor,

This study aims to investigate the main surface and sub-surface characteristics of several mesoscale eddies observed during eight Seaglider missions in the vicinity of Perth submarine canyon.

Although the topics and the applied methodology are promising, the paper reports several unclear or incomplete reasoning (sometimes caused by English mistakes and/or confused phrasing). The results are intriguing but at times appear superficially presented, which may challenge the reader's understanding. Additionally, I have concerns regarding the methodology used for eddy tracking, especially without further clarification from the authors on their rationale for this approach.

After a careful review of the paper, it presents as an interesting scientific work; however, it lacks the necessary attention to formal writing and presentation quality. Overall, it seems that different sections of the manuscript may have been written by various authors involved in the study, leading to inconsistencies in style. Despite this, the figures and statistical analyses are impressive, and their discussion is clear and well-articulated.

Therefore, I believe that this study will likely be a significant contribution after a careful (major) revision and clarification. I also suggest the grammar and the syntax to be better checked and verified to meet the high quality standards required for publication in Ocean Science. Specific comments/suggestions are listed in the attached document. Were a revised manuscript to be sent for another round of reviews, I would take part with pleasure.

ABSTRACT

The abstract maybe promises more than you actually find in the manuscript but it works fine. The studied region and the methodology is of huge interest for the oceanographic community and the paper seems exciting. So I was disappointed finding the manuscript a bit hurried.

Lines 8-10: Check and improve sentences

INTRODUCTION

Lines 26-32: Check syntax and grammar

Lines 54-55: Please improve this sentence

Figure 1: Eddy tracks often looks very "geometric", with sharp unrealistic changes in direction. I wonder if a different, more robust tracking algorithm can be used to verify (and eventually solve, if necessary) this issue.

Table 1: In caption, I would prefer to read "Adapted from ..."

DATA AND METHODS

Lines 122-126: In the first sentence I cannot get completely the meaning of your text (the word format is a typo or a verb is missing?). In the second one, I think that L3 is repeated more time than necessary. I suggest to improve product description rephrasing both sentences without repeating L3 several times.

Lines 128-133: Please reorganize this paragraph. I suggest to avoid repetitions and check the phrasing - NASA satellites, for example, do not distribute observations, but acquire. Data are not "applied" in SeaDAS but maybe "processed" and so on.

Lines 137-139: Is this safe? Any reference or sensitivity test about this scaling?

Line 141: I suggest to replace "see below" with "see section xxxx", and "corresponding" with "co-located".

Line 144: you already provided the symbol, you can use it avoiding to repeat relative vorticity

Line 152: As mentioned above, I wonder if the tracking algorithm proposed by Nencioli et al. (2010) is still the best option for your purposes, also considering that it was not developed for altimetry products. Three eddy detection and tracking methods (i.e., the Okubo–Weiss, vector-geometry, and winding-angle) algorithms can be usually applied for eddy identification and tracking with different performances as showed, for example, by Xing and Yang, 2021 (DOI: 10.1175/JTECH-D-20-0020.1). Then, several advanced techniques have been presented in the last two decades, also taking advantage of deep learning. In such a context, I would like you motivate your choice about Nencioli et al. (2010).

Lines 155-156: Cannot understand. Please rephrase.

Lines 156-159: Please avoid repetitions, for example merging the two sentences.

Line 170: "fitted", I would prefer "equipped"

Line 173: "In this study", I suggest to go to line 174 and start here a new paragraph.

Lines 181-182: I suggest to move this up, to line 174 –  "In this study, we analysed temperature, salinity, depth, and chlorophyll-a fluorescence observations collected during eight Seaglider missions carried out between October 2010 and January 2017 (Table 2), to study the vertical characteristics of mesoscale eddies".

Table 2: "Glider mission ID and total deployment period" sounds better for column 3

RESULTS

Line 189: KE was already defined in the previous section. If this represents the same parameter you can just use the acronym

Lines 203-204: What do you mean? Should not?

Line 204: "was characterized BY …" ?

Line 204: I suggest to avoid confusion using SSC and SCC - are they the same parameter? can you make a choice, or avoid acronym for SSC if not used anywhere than in the data section?

Figure 2: Can you improve the caption?
First sentence is very confusing and repetitive. Additionally, I cannot read the term "Prime" anywhere in the figure. For prime you mean the letter (not term) with the apostrophe?

Lines 217-218: "(For further … (2010))" - This is not necessary here in my opinion. You clearly mentioned it in the text. Again, I am not persuaded of using this tracking algorithm.

Figure 3: Please improve caption, especially in the first two sentences.

Lines 243-246: Please check this sentence: a comma or a pronoun is missing to clearly understand its meaning.

Line 251: "maximum" or "maxima" ? Also, please go through the text to homogenize formats, deciding if units must be attached or separate from numeric values.

Lines 253-256: Please, improve style in this important paragraph.

Line 257: Can you use a better term then "encompassing" here?

Lines 263 and 266: I suggest to avoid repeating similarly if possible.

Lines 269-271: Missing verb or wrong syntax. Please, improve.

Line 272: I don't get the meaning of this sentence. Please improve!

Lines 295-296: Respect to what ????

Lines 302-303: Please, improve this sentence. Verb, comma or pronoun is missing.

Figure 7: As for previous figures, please improve caption!

Line 344: better "from 18 to 21 March 2015"

Lines 359-361: Please improve this sentence.

Lines 362-364: Please improve.

Line 373: better "from 17 to 19 Nov in 2016"

Lines 388-389: Please check this sentence

DISCUSSION

Lines 400-401: Please, use commas!

Line 410: "over the period" reads better.

Lines 431-434: Please, improve these sentences.

Lines 445-446: Please improve, for example "The expected lower chlorophyll concentrations within anti-cyclonic eddies were not observed".

Line 451: Please use "to" or "with"

Lines 486-487: I suggest to rephrase as "ranged from ... to ..., and from ... to ..."

---

## Referee Comment (RC2)

Review for Preprint egusphere-2024-2901: Structure of mesoscale eddies in the vicinity of Perth Submarine Canyon

MAJOR COMMENT

This study presents an analysis of the surface and subsurface characteristics of mesoscale eddies observed during eight Seaglider missions near the Perth submarine canyon. The authors combine in-situ measurements collected between 2010 and 2017 and remote sensing data to describe the interactions with nine mesoscale eddies. Detailed descriptions of the eddies' physical characteristics, such as temperature, salinity, chlorophyll concentration, and mixed layer depth are provided, highlighting differences between eddy types and their potential impact on local oceanography.

The dataset itself is compelling and valuable, revealing the potential for intriguing patterns such as the absence of anticyclonic eddies above the shelf. However, the current manuscript is limited by its mostly descriptive approach, with minimal scientific analysis beyond basic characterization of eddy features. While the observational data are rich and has the potential to yield substantial insights, the authors have not fully utilized this potential to extract significant scientific findings. Without new and significant findings, the paper reads more as an overall data report rather than a scientific study.

I recommend that the authors incorporate some statistical analysis of the data, moving the manuscript's results beyond a descriptive focus and exploring the eddy dynamics and interactions quantitatively. Metrics on eddy intensity, variability (seasonal?), or nutrient entrainment would add a valuable quantitative dimension to the work. An analysis of the relationship between eddy characteristics (such as chlorophyll concentration and temperature profiles) and regional physical processes could provide new insights. Such analyses would allow for a clearer demonstration of the eddies' impact in the region, helping to elevate the manuscript from descriptive to analytical work. It is up to the authors to decide what processes they want to study, but new scientific findings are necessary to be included.

As such, I recommend the authors make major revisions to the paper, addressing these issues and with the expectation that the authors would focus their revision efforts on the inclusion of further scientific analysis. After the authors undertake these significant revisions, this study could become a noteworthy contribution to our understanding of mesoscale eddies in this unique region.

Some additional minor comments related to several grammar and syntax problems in the paper are provided below, along with some other specific suggestions in addition to the major comment above. The authors will need to have the text edited carefully, removing all grammatical/syntax errors, before it can be published. I would be happy to review an improved version of this paper if is submitted after editing.

MINOR COMMENTS:

Line 24: "oases" should be in quotations

Line 43: add a comma after "WA"

Line 66-67: You state that the paper's aim is to define the vertical structure of the eddies you measured by glider. I am not sure if this aim by itself is enough to justify publication of this work. Your aim should be to show something new related to these measurements, and shouldn't be limited to simple description of the data. As a reader, at this point I hope to find more than a mere description of the eddies characteristics in the upcoming sections.

Line 110-111: Something is strange about this sentence, please rephrase/edit.

Line 137-139: What is the likelihood of generating artifacts during the process of interpolation? More specifically, can you quantify the error introduced by detecting eddies onto a heavily interpolated field? Please address.

Line 149-150: Incomplete sentence - please check & edit.

Line 155-156: Sentence does not make sense, please rephrase.

Lines 191-192: It is difficult to gather this form Figure 1b. I can see that there is a collection of anticyclonic eddies closer to the coast, but it is hard form the figure to tell if this is the majority of them. Can you report the number of on-shelf AC eddies vs. Off-shelf AC eddies here? (The only thing I can really gather form figure 1b is the *absence* of cyclonic eddies near the coast.)

Line 200: You've mentioned "cyclonic/anti-cyclonic" elsewhere in the draft - better to keep with that terminology throughout the paper to refer to rotation direction.

Line 201: "(represented by yellow arrows in Figure 3)"

Lines 203-206: Not clear, there are several grammatical and description issues in these lines. Please rephrase & edit. Please have this paper revised and corrected for English writing.

Lines 431-433: Sentence is incomplete, please revise.

Lines 435-439: The fact that cyclonic eddies associated with upwelling didn't indicate higher chlorophyll in this case is a potentially interesting finding that should be explored further. This is an opportunity to go beyond the mere description of the data and include some findings.

Line 446: "Chlorophyll" (missing an L)

Lines 446-449: Please reword this sentence, the mechanism it describes needs to be made more clear.

Lines 451-452: Check grammar.

Lines 452-454: Again, some awkward grammatical description. To be clear, the entrainment of productive water was not impeded – rather, the direct observation of whether it occurred or not is what was impeded. Please rephrase to state correctly.

Looking at Figure 13, what I can see is a filament of high Chl at the upper edge of the eddy coming from the coast (panel c). I can see it to a much lesser degree in panel b, but less convincingly. In panel a, how can you be sure that is entrainement?

Lines 463-464: Check grammar.

Figure 1: Panel "b" in its current spaghetti plot format is not useful to the reader. The eddy tracks plotted like this are not discernible, and the figure hardly provides any insight into the actual behavior of the eddies. I suggest reworking this figure into a more discernible plot, perhaps by making use of the mean pathways/directions or by otherwise cleaning up the way the eddy tracks are displayed visually.

Figure 3: Check this description for grammar, it is not correct as written.

Figure 4: The way you've plotted these sections twice in the same figure (first from 0-900 m, then again from 0-500 m) is repetitive and unnecessary. Please choose only one depth range and eliminate the other (either 0-900 or 0-500). You can make better use of space for the figure that way and there will be no unneeded repetition. I leave it up to you how to reorganize the panels, but remove the repetition.

Figure 13: Check the colorbar on this figure: negative Chl values?

Figure 14: I do not think this figure helps. There is not much that can be gathered by this figure, except perhaps for some basic qualitative depictions. Aside from being shown an awkward scale and angle, it does not illustrate the characteristics of the eddies as you claim in the caption.

---

## Referee Comment (RC3)

Major comments :

- I recommend the authors to redefine the structure of the paper (for instance, having a more general and structured approach grouping cyclones and anticyclones in separate sections, as well as seasonal variability and impacts on chlorophyll-a) and to deepen their analysis of the mesoscale structures,

- The eddy tracking algorithm is relatively unsubstantial and not fully exploited. I am aware of the eddy detection algorithm by Nencioli et al. (2010). There is also a global atlas of eddy detection from gridded altimetry publicly available (eg https://www.aviso.altimetry.fr/en/data/products/value-added-products/global-mesoscale-eddy-trajectory-product.html). More recent detection methods such as AMEDA (Le Vu et al., 2018)) have also been widely used recently by the community working with satellite altimetry. So, I believe that there could be more appropriate tools or data sets to consider. More importantly, this might give a better description of eddy radius and intensity, as well as the possibility to better explore statistically the life-cycle of eddies formed in or passing by the study region (expanding the analysis of Figure 1). Please take the opportunity to revise the way mesoscale eddies are being detected, or improve the method description in light of existing methods.

- In line 143 where vorticity is defined, there are not enough details on how the Rossby number is calculated for each eddy. This part should be put after the eddy detection algorithm description, as it is closely linked and deduced from satellite currents. It should hence be mentioned how the Rossby number is calculated (average over a certain contour? and over time?). Since glider can also be used to characterize mesoscale eddies, I recommend the authors to go beyond the present analysis and try to compare the eddies characteristics seen from space with the ones inferred from their glider observations (as there can be noticeable differences, see for instance Yu et al. (2019))

Specific and minor comments :
- l8 : « particles » I would speak about water parcel more than particles for a fluid.
- l10 « as is »
- l15 and after : «anticlonic » without « - »
- l23 : « diameter >50km » : Mesoscale eddies haves scales ranging more generally from 10 to 100km depending on their location and water column stratification.
- l59 : Testor et al. (2018) could be cited here.
- l62 : Bosse et al. (2016) could also be cited here.
- l94 : what is the inertial frequency in the study area ? Please rephrase the sentence ending with « the resonant wind driven inertial currents extend to 500m water depth », there is something wrong with it.
- l102 : The generation of eddies in that context is actually very particular. What about baroclinic/barotropic instability ?
- L128 : « distributed by »
- section 2.3 : there should be details on glider data processing like thermal lag correction and non-photochemical quenching for chlorophyll-a.
- l168 : « m » not « km »
- l202 : « Rossby number, Ro » (Ro, no need for subscript)
- l203 and 462 : I don't think that the conclusion about the Coriolis frequency is right. Rossby number indicates whether the eddy dynamics is in geostrophic balance (Ro<<1) or cyclogeostrophic balance when Ro approach unity.
- Figure 4 and others : There are interpolation artefact showing triangular shapes at the bottom of each glider sections. Please consider another way of representing glider sections.
- l400-410 : this feels like an introduction part...
- Fugure 14 is not very informative and rather confusing with the 3D perspective and different scales invovled.

References :
- Le Vu, B., Stegner, A., & Arsouze, T. (2018). Angular momentum eddy detection and tracking algorithm (AMEDA) and its application to coastal eddy formation. Journal of Atmospheric and Oceanic Technology, 35(4), 739-762.
- Testor, P., De Young, B., Rudnick, D. L., Glenn, S., Hayes, D., Lee, C. M., ... & Wilson, D. (2019). OceanGliders: a component of the integrated GOOS. Frontiers in Marine Science, 6, 422.
- Bosse, A., Testor, P., Houpert, L., Damien, P., Prieur, L., Hayes, D., ... & Mortier, L. (2016). Scales and dynamics of S ubmesoscale C oherent V ortices formed by deep convection in the northwestern M editerranean S ea. Journal of Geophysical Research: Oceans, 121(10), 7716-7742.
- Yu, L. S., Bosse, A., Fer, I., Orvik, K. A., Bruvik, E. M., Hessevik, I., & Kvalsund, K. (2017). The L ofoten B asin eddy: Three years of evolution as observed by Seagliders. Journal of Geophysical Research: Oceans, 122(8), 6814-6834.

---

## Author Comment (AC1)

**Author's Response for RC1**

This study aims to investigate the main surface and sub-surface characteristics of several mesoscale eddies observed during eight Seaglider missions in the vicinity of Perth submarine canyon. Although the topics and the applied methodology are promising, the paper reports several unclear or incomplete reasoning (sometimes caused by English mistakes and/or confused phrasing). The results are intriguing but at times appear superficially presented, which may challenge the reader's understanding. Additionally, I have concerns regarding the methodology used for eddy tracking, especially without further clarification from the authors on their rationale for this approach. After a careful review of the paper, it presents as an interesting scientific work; however, it lacks the necessary attention to formal writing and presentation quality. Overall, it seems that different sections of the manuscript may have been written by various authors involved in the study, leading to inconsistencies in style. Despite this, the figures and statistical analyses are impressive, and their discussion is clear and well articulated. Therefore, I believe that this study will likely be a significant contribution after a careful (major) revision and clarification. I also suggest the grammar and the syntax to be better checked and verified to meet the high quality standards required for publication in Ocean Science. Specific comments/suggestions are listed in the attached document. Were a revised manuscript to be sent for another round of reviews, I would take part with pleasure.

**Response:** We sincerely thank the reviewer for their thoughtful feedback and constructive comments. We acknowledge the concerns regarding inconsistencies in writing style, unclear phrasing, and the methodology for eddy tracking. These have been rectified in the revised paper. The Nencioli et al. (2010) algorithm as it has been used by the UWA research group with good outcomes (Bitencourt et al., 2024; Cosoli, et al., 2020; Kodithuwakku 2024). It should also be noted that the emphasis of the paper is on the vertical structure of the eddies as sampled through ocean gliders. The eddy detection algorithm was used only demonstrate eddy activity within the Perth canyon, basically to construct Figure 1b. As it does not influence the results/discussion we have removed it from the revised version of the manuscript.

In the revised manuscript, we have improved the grammar, syntax, and overall presentation ensuring a more coherent and professional narrative. We appreciate the positive feedback on the figures and statistical analyses, and we will ensure consistency in style and tone throughout the manuscript. We are

grateful for the reviewer's time, and we are confident that the revisions will strengthen the manuscript. A point-by-point response to the specific comments are provided with the revised manuscript.

ABSTRACT

The abstract maybe promises more than you actually find in the manuscript but it works fine. The studied region and the methodology is of huge interest for the oceanographic community and the paper seems exciting. So I was disappointed finding the manuscript a bit hurried.

**Response:** Thank you for the reviewer's suggestion. We have revised the abstract substantially and believe that the revised abstract address the shortcomings.

Lines 8-10: Check and improve sentences

**Response:** The sentence has been revised in accordance with the comment.

INTRODUCTION

Lines 26-32: Check syntax and grammar

**Response:** The sentences have been revised in accordance with the comment.

Lines 54-55: Please improve this sentence

**Response:** The sentence has been deleted as it does not add any additional information.

Figure 1: Eddy tracks often looks very "geometric", with sharp unrealistic changes in direction. I wonder if a different, more robust tracking algorithm can be used to verify (and eventually solve, if necessary) this issue.

**Response:** As per general comment – we have omitted Figure 1b and Nencioli method completely.

Table 1: In caption, I would prefer to read "Adapted from ..."

**Response:** The caption has been revised in accordance with the comment.

DATA AND METHODS

Lines 122-126: In the first sentence I cannot get completely the meaning of your text (the word format is a typo or a verb is missing?). In the second one, I think that L3 is repeated more time than necessary. I suggest to improve product description rephrasing both sentences without repeating L3 several times.
**Response:** We have revised the whole section for better clarity and reading.

Lines 128-133: Please reorganize this paragraph. I suggest to avoid repetitions and check the phrasing -NASA satellites, for example, do not distribute observations, but acquire. Data are not "applied" in SeaDAS but maybe "processed" and so on.
**Response:** We have revised the whole section for better clarity and reading.

Lines 137-139: Is this safe? Any reference or sensitivity test about this scaling?
**Response:** The scaling used in lines 137-139 is supported by sensitivity test results undertaken by Kodithuwakku (2024) an optimal resolution of 0.05° (5.5 km) is effective for detecting eddy features. However, this comment is irrelevant as eddy detection has been removed from the manuscript.

Line 141: I suggest to replace "see below" with "see section xxxx", and "corresponding" with "co-located".
**Response:** The sentences have been revised in accordance with the comment.

Line 144: you already provided the symbol, you can use it avoiding to repeat relative vorticity
**Response:** The sentence has been revised with symbol in accordance with the comment.

Line 152: As mentioned above, I wonder if the tracking algorithm proposed by Nencioli et al. (2010) is still the best option for your purposes, also considering that it was not developed for altimetry products. Three eddy detection and tracking methods (i.e., the Okubo–Weiss, vector-geometry, and winding-angle) algorithms can be usually applied for eddy identification and tracking with different

performances as showed, for example, by Xing and Yang, 2021 (DOI: 10.1175/JTECH-D-20-0020.1). Then, several advanced techniques have been presented in the last two decades, also taking advantage of deep learning. In such a context, I would like you motivate your choice about Nencioli et al. (2010).

**Response:** As mentioned above we have omitted Figure 1b and no longer require eddy tracking. The emphasis of the paper is on the vertical structure of the eddies.

Lines 155-156: Cannot understand. Please rephrase.

**Response:** The sentence has been revised in accordance with the comment.

Lines 156-159: Please avoid repetitions, for example merging the two sentences.

**Response:** The sentences have been merged to avoid repetition as per the comment.

Line 170: "fitted", I would prefer "equipped"

**Response:** The sentence has been revised in accordance with the comment.

Line 173: "In this study", I suggest to go to line 174 and start here a new paragraph.

**Response:** A new paragraph has been started from line 174 as per the comment.

Lines 181-182: I suggest to move this up, to line 174 – "In this study, we analysed temperature, salinity, depth, and chlorophyll-a fluorescence observations collected during eight Seaglider missions carried out between October 2010 and January 2017 (Table 2), to study the vertical characteristics of mesoscale eddies".

**Response:** The sentence has been moved to line 174 as per the comment.

Table 2: "Glider mission ID and total deployment period" sounds better for column 3

**Response:** The column title has been revised as per the comment.

RESULTS

Line 189: KE was already defined in the previous section. If this represents the same parameter you can just use the acronym

**Response:** The sentence has been revised with acronym in accordance with the comment.

Lines 203-204: What do you mean? Should not?

**Response:** The phrase "This indicated that the Coriolis force was important in the eddy dynamics" was meant to highlight that the calculated Rossby numbers reflect the dominance of the Coriolis force in both cyclonic and anticyclonic eddies. The sentences have been updated for clarity.

Line 204: "was characterized BY …" ?

**Response:** The sentence has been revised in accordance with the comment.

Line 204: I suggest to avoid confusion using SSC and SCC - are they the same parameter? can you make a choice, or avoid acronym for SSC if not used anywhere than in the data section?

**Response:** The sentences have been revised to use 'Sea Surface Chlorophyll' (SSC) while avoiding the use of SCC.

Figure 2: Can you improve the caption?

First sentence is very confusing and repetitive. Additionally, I cannot read the term "Prime" anywhere in the figure. For prime you mean the letter (not term) with the apostrophe?

**Response:** The caption has been revised as per the comment. The term "prime" has been clarified to indicate the letter with an apostrophe, aligning with the figure labelling.

Lines 217-218: "(For further … (2010))" - This is not necessary here in my opinion. You clearly mentioned it in the text. Again, I am not persuaded of using this tracking algorithm.

**Response:** The figure 1b has been removed.

Figure 3: Please improve caption, especially in the first two sentences.

**Response:** The caption has been revised in accordance with the comment.

Lines 243-246: Please check this sentence: a comma or a pronoun is missing to clearly understand its

meaning.

**Response:** The sentence has been revised by separating the ideas into two sentences and ensuring proper punctuation.

Line 251: "maximum" or "maxima" ? Also, please go through the text to homogenize formats, deciding if units must be attached or separate from numeric values.

**Response:** The term "maximum" has been held as it refers to a single peak value rather than multiple peaks. Also, the format has been updated with units separated from numeric values where appropriate.

Lines 253-256: Please, improve style in this important paragraph.

**Response:** The sentences has been revised in accordance with the comment.

Line 257: Can you use a better term then "encompassing" here?

**Response:** The term "encompassing" has been replaced with "including" to enhance clarity.

Lines 263 and 266: I suggest to avoid repeating similarly if possible.

**Response:** The repetition of "similarly" has been avoided by restructuring the sentences.

Lines 269-271: Missing verb or wrong syntax. Please, improve.

**Response:** The sentences has been revised in accordance with the comment.

Line 272: I don't get the meaning of this sentence. Please improve!

**Response:** The sentences has been revised in accordance with the comment.

Lines 295-296: Respect to what ????

**Response:** The sentence has been revised to clarify that the observed characteristics are relative to the surrounding waters.

Lines 302-303: Please, improve this sentence. Verb, comma or pronoun is missing.

**Response:** The sentences has been revised in accordance with the comment.

Figure 7: As for previous figures, please improve caption!

**Response:** Figure 7 differs from the previous figures in content, so the caption has been kept as it is to maintain clarity.

Line 344: better "from 18 to 21 March 2015"

**Response:** The sentences has been revised in accordance with the comment.

Lines 359-361: Please improve this sentence.

**Response:** The sentences has been revised in accordance with the comment.

Lines 362-364: Please improve.

**Response:** The sentences has been refined to enhance the clarity.

Line 373: better "from 17 to 19 Nov 2016"

**Response:** The sentences has been revised in accordance with the comment.

Lines 388-389: Please check this sentence

**Response:** The sentence has been revised to improve clarity.

DISCUSSION

Lines 400-401: Please, use commas!

**Response:** The sentence has been revised in accordance with the comment.

Line 410: "over the period" reads better.

**Response:** The sentence has been revised in accordance with the comment.

Lines 431-434: Please, improve these sentences.

**Response:** The sentences have been revised in accordance with the comment.

Lines 445-446: Please improve, for example "The expected lower chlorophyll concentrations within anticyclonic eddies were not observed".

**Response:** The sentence has been revised in accordance with the comment.

Line 451: Please use "to" or "with"

**Response:** The sentence has been revised in accordance with the comment.

Lines 486-487: I suggest to rephrase as "ranged from ... to ..., and from ... to ..."

**Response:** The sentence has been revised in accordance with the comment.

---

## Author Comment (AC3)

**Author's Response for RC2**

MAJOR COMMENT

This study presents an analysis of the surface and subsurface characteristics of mesoscale eddies observed during eight Seaglider missions near the Perth submarine canyon. The authors combine in-situ measurements collected between 2010 and 2017 and remote sensing data to describe the interactions with nine mesoscale eddies. Detailed descriptions of the eddies' physical characteristics, such as temperature, salinity, chlorophyll concentration, and mixed layer depth are provided, highlighting differences between eddy types and their potential impact on local oceanography.

The dataset itself is compelling and valuable, revealing the potential for intriguing patterns such as the absence of anticyclonic eddies above the shelf. However, the current manuscript is limited by its mostly descriptive approach, with minimal scientific analysis beyond basic characterization of eddy features. While the observational data are rich and has the potential to yield substantial insights, the authors have not fully utilized this potential to extract significant scientific findings. Without new and significant findings, the paper reads more as an overall data report rather than a scientific study.

I recommend that the authors incorporate some statistical analysis of the data, moving the manuscript's results beyond a descriptive focus and exploring the eddy dynamics and interactions quantitatively. Metrics on eddy intensity, variability (seasonal?), or nutrient entrainment would add a valuable quantitative dimension to the work. An analysis of the relationship between eddy characteristics (such as chlorophyll concentration and temperature profiles) and regional physical processes could provide new insights. Such analyses would allow for a clearer demonstration of the eddies' impact in the region, helping to elevate the manuscript from descriptive to analytical work. It is up to the authors to decide what processes they want to study, but new scientific findings are necessary to be included.

As such, I recommend the authors make major revisions to the paper, addressing these issues and with the expectation that the authors would focus their revision efforts on the inclusion of further scientific analysis. After the authors undertake these significant revisions, this study could become a noteworthy contribution to our understanding of mesoscale eddies in this unique region.

Some additional minor comments related to several grammar and syntax problems in the paper are provided below, along with some other specific suggestions in addition to the major comment above. The authors will need to have the text edited carefully, removing all grammatical/syntax errors, before

it can be published. I would be happy to review an improved version of this paper if is submitted after editing.

**Response:** We sincerely thank the reviewer for their detailed and constructive feedback on our manuscript. We appreciate the recognition of the value and potential of our dataset, as well as the acknowledgment of the findings it reveals. The importance of moving beyond a descriptive focus to provide a more analytical exploration of mesoscale eddy dynamics and their impacts is acknowledged. However, accessing only field measurements using gliders limits the ability to undertake additional analysis. A companion paper, which is currently under review, has included more detailed statistics on eddy variability. Minor comments regarding grammar and syntax have been acknowledged, and these issues have been carefully addressed in the revised manuscript. To ensure the writing meets the high standards required for publication, a thorough review of the text has been conducted. Confidence has been gained that the revisions have significantly enhanced the quality and impact of the work. The reviewer's encouragement and willingness to review a revised version of the manuscript have been sincerely appreciated. A point-by-point response to all comments, including specific suggestions and minor corrections, has been provided with the revised submission.

MINOR COMMENTS:

Line 24: "oases" should be in quotations

**Response:** The sentence has been revised in accordance with the comment.

Line 43: add a comma after "WA"

**Response:** The sentence has been revised to include a comma after 'WA,' as per the comment.

Line 66-67: You state that the paper's aim is to define the vertical structure of the eddies you measured by glider. I am not sure if this aim by itself is enough to justify publication of this work. Your aim should be to show something new related to these measurements, and shouldn't be limited to simple description of the data. As a reader, at this point I hope to find more than a mere description of the eddies characteristics in the upcoming sections.

**Response:** The sentence has been revised to reflect a more focused approach on uncovering new findings related to eddy dynamics and their role in the ecosystem. One of the major findings is the higher SSC in anti-cyclonic eddies due to entrainment of higher SSC water from the continental shelf.

Line 110-111: Something is strange about this sentence, please rephrase/edit.
**Response:** The sentence has been revised in accordance with the comment.

Line 137-139: What is the likelihood of generating artifacts during the process of interpolation? More specifically, can you quantify the error introduced by detecting eddies onto a heavily interpolated field? Please address.
**Response:** This is not relevant as we have removed the eddy detection section from the manuscript.

Line 149-150: Incomplete sentence - please check & edit.
**Response:** The sentence has been revised for clarity.

Line 155-156: Sentence does not make sense, please rephrase.
**Response:** The sentence has been revised in accordance with the comment.

Lines 191-192: It is difficult to gather this form Figure 1b. I can see that there is a collection of anticyclonic eddies closer to the coast, but it is hard form the figure to tell if this is the majority of them. Can you report the number of on-shelf AC eddies vs. Off-shelf AC eddies here? (The only thing I can really gather form figure 1b is the absence of cyclonic eddies near the coast.)
**Response:** This is not relevant as we have removed Figure 1b the manuscript. We agree that it does not contribute to the aims of the manuscript.

Line 200: You've mentioned "cyclonic/anti-cyclonic" elsewhere in the draft - better to keep with that terminology throughout the paper to refer to rotation direction.
**Response:** The sentences have been revised throughout the paper to consistently use "cyclonic/anticyclonic".

Line 201: "(represented by yellow arrows in Figure 3)"

**Response:** The sentence has been revised in accordance with the comment.

Lines 203-206: Not clear, there are several grammatical and description issues in these lines. Please rephrase & edit. Please have this paper revised and corrected for English writing.

**Response:** The sentences have been revised to address the grammatical and descriptive issues.

Lines 431-433: Sentence is incomplete, please revise.

**Response:** The sentence has been revised in accordance with the comment.

Lines 435-439: The fact that cyclonic eddies associated with upwelling didn't indicate higher chlorophyll in this case is a potentially interesting finding that should be explored further. This is an opportunity to go beyond the mere description of the data and include some findings.

**Response:** We thank the reviewer for highlighting the interesting observation regarding chlorophyll concentrations. This finding has been recognized as a potentially significant deviation from the usual paradigm. In the revised manuscript, have expanded the discussion to highlight the implications of these findings.

Line 446: "Chlorophyll" (missing an L)

**Response:** The sentence has been revised in accordance with the comment.

Lines 446-449: Please reword this sentence, the mechanism it describes needs to be made more clear.

**Response:** The revised wording has been updated to provide a clearer explanation of how the higher chlorophyll concentrations in anticyclonic eddies result from the entrainment of chlorophyll-rich shelf waters, which are transported offshore as the eddy moves westward-see also new Figure 14c.

Lines 451-452: Check grammar.

**Response:** The sentence has been revised for clarity and grammatical accuracy.

Lines 452-454: Again, some awkward grammatical description. To be clear, the entrainment of productive water was not impeded – rather, the direct observation of whether it occurred or not is what was impeded. Please rephrase to state correctly.

**Response:** The wording has been revised for clarity in accordance with the comment.

Looking at Figure 13, what I can see is a filament of high Chl at the upper edge of the eddy coming from the coast (panel c). I can see it to a much lesser degree in panel b, but less convincingly. In panel a, how can you be sure that is entrainment?

Lines 463-464: Check grammar.

**Response:** The sentence has been revised for grammatical accuracy.

Figure 1: Panel "b" in its current spaghetti plot format is not useful to the reader. The eddy tracks plotted like this are not discernible, and the figure hardly provides any insight into the actual behavior of the eddies. I suggest reworking this figure into a more discernible plot, perhaps by making use of the mean pathways/directions or by otherwise cleaning up the way the eddy tracks are displayed visually.

**Response:** We have removed figure 1b.

Figure 3: Check this description for grammar, it is not correct as written.

**Response:** The description of Figure 3 has been revised for grammatical accuracy in accordance with the comment.

Figure 4: The way you've plotted these sections twice in the same figure (first from 0-900 m, then again from 0-500 m) is repetitive and unnecessary. Please choose only one depth range and eliminate the other (either 0-900 or 0-500). You can make better use of space for the figure that way and there will be no unneeded repetition. I leave it up to you how to reorganize the panels, but remove the repetition.

**Response:** Both depth ranges (0-900 m and 0-500 m) have been retained to provide clarity and support the description in the text, as this has been intended to highlight different aspects of the data. This approach has been chosen to ensure the entire data record across the transect is visible while allowing for a more detailed focus on the upper water column.

Figure 13: Check the colorbar on this figure: negative Chl values?

**Response:** The colorbar in Figure 13 has been presented in logarithmic scale, which is why it includes negative values. This has been clarified in the caption.

Figure 14: I do not think this figure helps. There is not much that can be gathered by this figure, except perhaps for some basic qualitative depictions. Aside from being shown an awkward scale and angle, it does not illustrate the characteristics of the eddies as you claim in the caption.

**Response:** The figure has been revised to address the concerns raised.